# The Function of the Mutant p53-R175H in Cancer

**DOI:** 10.3390/cancers13164088

**Published:** 2021-08-13

**Authors:** Yen-Ting Chiang, Yi-Chung Chien, Yu-Heng Lin, Hui-Hsuan Wu, Dung-Fang Lee, Yung-Luen Yu

**Affiliations:** 1Graduate Institute of Biomedical Sciences, China Medical University, Taichung 40402, Taiwan; u104001403@cmu.edu.tw (Y.-T.C.); chenyc0710@mail.cmu.edu.tw (Y.-C.C.); u109210209@cmu.edu.tw (Y.-H.L.); u108210214@cmu.edu.tw (H.-H.W.); 2Program for Translational Medicine, China Medical University, Taichung 40402, Taiwan; 3Institute of New Drug Development, China Medical University, Taichung 40402, Taiwan; 4Drug Development Center, Research Center for Cancer Biology, China Medical University, Taichung 40402, Taiwan; 5Center for Molecular Medicine, China Medical University Hospital, Taichung 40402, Taiwan; 6Department of Integrative Biology and Pharmacology, McGovern Medical School, The University of Texas Health Science Center at Houston, Houston, TX 77030, USA; 7UTHealth Graduate School of Biomedical Sciences, The University of Texas MD Anderson Cancer Center, Houston, TX 77030, USA; 8Center for Stem Cell and Regenerative Medicine, The Brown Foundation Institute of Molecular Medicine for the Prevention of Human Diseases, The University of Texas Health Science Center at Houston, Houston, TX 77030, USA; 9Center for Precision Health, School of Biomedical Informatics and School of Public Health, The University of Texas Health Science Center at Houston, Houston, TX 77030, USA; 10Department of Medical Laboratory Science and Biotechnology, Asia University, Taichung 41354, Taiwan

**Keywords:** mutant p53, p53 R175H, gain-of-function, targeted therapy, immunotherapy, cancer

## Abstract

**Simple Summary:**

*TP53* is one of the most well-known and intensively studied tumor-suppressor genes. *TP53* is also the most commonly mutated gene in cancer. Many *TP53* mutations are missense mutations and are located in several hotspots. Increasing evidence has shown that these hotspot mutations both lose the wild-type function and gain oncogenic functions to promote cancer progression. Among these hotspot mutations, p53-R175H has the highest occurrence in diverse cancers. In this review, we summarize studies associated with p53-R175H gain of function, and outline the current situation of the development of small molecules or immunotherapies that target p53-R175H.

**Abstract:**

Wild-type p53 is known as “the guardian of the genome” because of its function of inducing DNA repair, cell-cycle arrest, and apoptosis, preventing the accumulation of gene mutations. *TP53* is highly mutated in cancer cells and most *TP53* hotspot mutations are missense mutations. Mutant p53 proteins, encoded by these hotspot mutations, lose canonical wild-type p53 functions and gain functions that promote cancer development, including promoting cancer cell proliferation, migration, invasion, initiation, metabolic reprogramming, angiogenesis, and conferring drug resistance to cancer cells. Among these hotspot mutations, p53-R175H has the highest occurrence. Although losing the transactivating function of the wild-type p53 and prone to aggregation, p53-R175H gains oncogenic functions by interacting with many proteins. In this review, we summarize the gain of functions of p53-R175H in different cancer types, the interacting proteins of p53-R175H, and the downstream signaling pathways affected by p53-R175H to depict a comprehensive role of p53-R175H in cancer development. We also summarize treatments that target p53-R175H, including reactivating p53-R175H with small molecules that can bind to p53-R175H and alter it into a wild-type-like structure, promoting the degradation of p53-R175H by targeting heat-shock proteins that maintain the stability of p53-R175H, and developing immunotherapies that target the p53-R175H–HLA complex presented by tumor cells.

## 1. Introduction

Cancer is a set of diseases characterized by abnormal cell growth with the potential to invade other parts of the body. Douglas Hanahan and Robert A. Weinberg proposed six hallmarks of cancer: sustaining proliferative signaling, evading growth suppressors, resisting cell death, enabling replicative immortality, inducing angiogenesis, and activating invasion and metastasis [1]. These hallmarks can be acquired by mutations of genes that are involved in these biological processes. p53, a sequence-transcription factor encoded by the *TP53* gene, is called “the guardian of the genome” because it can sense DNA damage in cells and turn on its downstream signaling that can induce DNA repair, cell-cycle arrest, and apoptosis by modulating the expression of its target genes. Once the p53 pathway has been altered, genetically damaged cells will not go into senescence or apoptosis, causing the accumulation of mutations and acquiring hallmarks of cancer [2]. The p53 protein has an N-terminus containing transactivation domains, a DNA binding domain that binds to specific DNA sequences, and a C-terminus oligomerization and regulatory domains [3]. In data from The Cancer Genome Atlas (TCGA) program, 41.8% of cancer patients have alterations in *TP53*, suggesting that *TP53* is highly mutated in cancer patients. Most *TP53* mutations are located within the DNA binding domain (96–293 aa) and at several hotspots, such as R175, R248, R273, and R282 (Figure 1a). The pattern of *TP53* hotspots distribution from the International Agency for Research on Cancer (IARC) *TP53* somatic mutation database is similar to the TCGA database. R175, G245, R248, R249, R273, and R282 are the top six frequently mutated sites (Figure 1b). As for the IARC *TP53* germline mutation database, the distribution of hotspots is similar to the somatic database, except for the high mutation frequency in R337 (Figure 1c) [4,5]. *TP53-R337H*, also known as “the Brazilian germline *TP53* mutation”, increases predisposition to childhood adrenocortical tumor, while the detailed molecular mechanism needs further investigation [6]. There might be several reasons that could explain why there are hotspot mutations in the *TP53* gene. First, different mutations might have different levels of impact on the function of p53, either by altering the global protein structure or disrupting the p53–DNA interface, and there is a selection for cancer cells that express p53 mutants, losing its wild-type function [7]. On the basis of the crystal structure of the p53–DNA complex [8], hotspots R248 and R273 come into direct contact with DNA, whereas R249 and R282 lie near the DNA contacting interface. The R175 hotspot is located at the zinc-binding site near the DNA binding interface, which is very important to the maintenance of structural stability. Therefore, the p53-R175H mutation causes a change in the protein structure. Based on the crystal structure, these hotspots can be classified as contact mutants (R248 and R273) which make direct contact with DNA and structural mutants (R175, G245, R249, and R282) which maintain the structure of the DNA binding interface [8]. Second, the occurrence of C to T transitions increased 10-fold in the 5′ methylated cytosine of CpG codons. Four of the six codons encoding arginine have CpG dinucleotides in the first two positions, which can explain why the top six most frequent missense mutations all occur at the arginine residue [9]. Lastly, missense mutations located at these hotspots encode mutant p53 oncoproteins that lose canonical tumor-suppressor functions and gain oncogenic functions to promote cancer progression. Among these hotspot mutations, p53-R175H has the highest occurrence in cancer patients in both the TCGA database and IARC *TP53* somatic mutations database (Table 1). The p53-R175H loses the transactivating function of the wild-type p53 due to its defective DNA binding interface. However, p53-R175H gains function by many kinds of mechanisms. It is reported that p53-R175H can bind to some DNA sequences which are different from the wild-type p53 response element and transactivate its target genes [10]. p53-R175H can also interact with numerous kinds of transcription factors to enhance or suppress the expression of their target genes [11]. In addition, the p53-R175H is more prone to aggregation than the wild-type p53 because it exhibits a larger hydrophobic surface area and higher loop flexibility than the wild-type p53 [12]. p53-R175H aggregates can not only induce coaggregation of wild-type p53 to cause the loss of function but also induce coaggregation of p63 and p73 to cause gain-of-functions [13]. Different p53 mutants may gain different functions. In this review, we focus on the gain-of-function of p53-R175H and discuss its role in cancer development. In the last section, we summarize the current development of mutant-p53-targeted therapies.

## 2. Gain-of-Function of Mutant *p53-R175H*

A large part of *TP53* mutations in cancer cells is missense mutations which encode altered proteins, suggesting that these altered proteins may contribute some functions and are not just the result of loss of function mutations. In 1993, Dittmer et al. reported that mutant p53 proteins (V143A, R175H, R248W, R273H, and D281G) expressed in murine fibroblast cell lines ((10)3 cells) lacking p53 resulted in enhanced tumorigenic potential in nude mice. Mutant p53 alleles can also enhance the plating efficiency of the human osteosarcoma cell lines SAOS-2 and enhance the expression of the *CAT* gene in (10)3 cells [14]. In 2004, Olive et al. reported that p53^R172H/−^ (mouse R172H refers to human R175H) and p53^R270H/−^ (mouse R270H refers to human R273H) mice developed novel tumors compared to p53^−/−^ mice, including a variety of carcinomas and more frequent endothelial tumors [15]. These two studies provided strong evidence to the mutant p53 gain-of-function hypothesis. In the last two decades, more and more studies have shown that p53 gain-of-function plays important role in tumorigenesis, and the mechanism has been investigated. Mutant p53 can modulate the expression of some genes by binding to their promotor with or without the help of different transcription factors, turning on many downstream oncogenic pathways. Mutant p53 can also bind to some enzymes to gain new functions. In this chapter, we summarize the p53-R175H interacting proteins, target genes, and gain-of-function. We also list other p53 mutants that have gain-of-function the same as p53-R175H in each study to make it easy to find which p53 mutants have phenotypes similar to p53-R175H or which molecular mechanisms are p53-R175H specific.

### 2.1. Inducing Genetic Instability

Acquisition of the multiple cancer hallmarks depends on a succession of alterations in the genomes of neoplastic cells [1]. *TP53* is the central of the genomic integrity maintaining system because it can sense stress signals and force genetically damaged cells into either senescence or apoptosis [2]. However, it was reported that p53-R175H gains the function of inducing genetic instability. Hingorani et al. generated mouse pancreatic ductal adenocarcinoma model containing *Trp53^R172H^* and *Kras^G12D^*. They found that the pancreatic carcinoma cells from mice expressing *Trp53^R172H^* frequently contain greater than two centrosomes while the pancreatic carcinoma cells from *Trp53* wild-type mice have a normal number of centrosomes [16]. Liu et al. further identified that p53-R175H mice are cancer prone and in mouse embryonic fibroblast p53-R175H impairs the recruitment of MRN/ATM to DNA double strain break sites by interacting with MRN complex component Mre11, thereby inducing chromosomal translocation [17] (Table 2).

### 2.2. Promoting Tumor Cell Growth

Since the *TP53* mutation often occurs in several hotspots by single nucleotide substitution resulting in missense mutation, scientists considered whether these mutant *p53* proteins both lose the wild-type p53 function and gain some unique functions to promote cancer-cell growth. In 2004, Scian et al. expressed hotspot mutants p53-R175H, p53-R273H, and p53-D281G in H1299, which is a p53 null human non-small-cell lung carcinoma cell line, and found that the proliferation rate of these cells increased compared to that of control cells. They also found that some growth-related genes, which are not the target genes of wild-type p53, are upregulated by these p53 mutants. However, they did not investigate the detailed mechanism [18]. *GEF-H1*, a guanine exchange factor-H1 for RhoA, was proved to be a p53-R175H, p53-V157F, and p53-R248Q target gene. The growth of many kinds of mutant p53 cancer cell lines but not the wild-type p53 cell lines is dependent on the expression of GEF-H1 [19]. Growth-regulated oncogene 1 (*GRO1* or *CXCL1*) was reported to be another p53-R175H target gene. p53-R175H can transactivate the *GRO1* gene by directly binding to its promoter region. Knockdown of GRO1 in colon adenocarcinoma cell line SW480 and pancreatic ductal adenocarcinoma cell line MIA PaCa-2 decreases cell viability [20]. In 2015, Zhu et al. reported a novel mutant *p53* gain-of-function mechanism that is associated with the chromatin pathway [21]. p53 with hotspot mutation can transcriptionally upregulate chromatin regulatory genes, including methyltransferases *MLL1* (*KMT2A*), *MLL2* (*KMT2D*), and acetyltransferase *MOZ* (*KAT6A*) by binding to their promoter regions together with ETS2, which is an ETS family transcription factor. Upregulation of these methyltransferases and acetyltransferase results in global increases in histone methylation and acetylation, and alters the expression of many genes that are essential to tumor growth. TCGA database analysis also revealed that, in many cancer types, *MLL1*, *MLL2*, and *MOZ* gene expression are upregulated in tumors with p53 hotspot mutation (R175H, R248Q, R248W, R249S, and R273H) compared to tumors with wild-type p53 [21]. This finding underscored that mutant p53 can alter the whole transcriptome by regulating epigenetic gene expression. In breast-cancer cell lines, mutant p53s can also bind to transcription factor NF-Y and cofactor YAP, and together activate several genes associated with cell growth, including *cyclin A*, *cyclin B*, and *CDK1* genes [22]. Additionally, p53-R175H and p53-R273H can cooperate with transcription factor MYB and transactivate replication-initiation-related genes *CDC7* and *DBF4.* High *CDC7* expression correlates with mutant *p53* expression and poor prognosis in lung-adenocarcinoma patients [23]. Furthermore, Valentino et al. found that, in breast and prostate cancers, p53-R175H and p53-R280K can induce PI3K/AKT signaling pathway and promote tumor growth by binding to tumor suppressor DAB2IP. DAB2IP can directly bind to PI3K and AKT, and limit their activation [24]. Pancreatic cancer is driven by *KRAS* and *TP53* mutations. However, we knew little about the relationship between these two genes. Recently, Escobar-Hoyos et al. discovered that p53-R175H increases the expression of *hnRNPK*, which is a splicing regulator, resulting in the production of isoforms of GTPase-activating proteins (GAPs). These GAP isoforms lost their function to contact and inactivate mutant KRAS, so KRAS downstream signaling pathways are turned on and contribute to tumor growth [25]. All these findings suggest that p53-R175H gains the function of promoting tumor-cell growth in many kinds of cancer types and by different kinds of molecular mechanisms (Table 3). 

### 2.3. Promoting Tumor Migration, Invasion, and Metastasis

p53-R175H promotes tumor growth and increases its mobility. In 2000, Liu et al. first observed this phenomenon in heterozygous p53^R172H^ mice (mouse R172H refers to human R175H) [26]. They generate mice containing p53^R172H/+^. Comparing to p53^−/+^ mice, p53^R172H/+^ mice have a higher frequency to develop carcinomas and osteosarcoma, and these carcinomas and osteosarcoma also have a high frequency to metastasize. However, osteosarcoma developed in p53^−/+^ mice rarely metastasize. This result implied that p53-R175H gains function in promoting tumor metastasis [26]. Morton et al. also found that p53-R175H promoted tumor metastasis in the *KRAS*/*TP53* mutation pancreatic ductal adenocarcinoma (PDAC) mouse model. p53-R172H mice have similar potential to develop PDAC comparing to p53-null mice. However, only tumor cells containing p53-R172H have the invasive ability [27]. Weissmueller et al. reported that mutant p53 binds to tumor suppressor p73 and prevents p73 binding to the NF-YA, NF-YB, and NF-YC complex. Therefore, the NF-Y complex can transactivate PDGFRb, which is an essential gene to pancreatic cancer cell invasion [28]. p73 shows a high structural resemblance to p53. TAp73, an isoform of p73, can serve as a tumor suppressor by inducing DNA repair, cell-cycle arrest, and apoptosis [29]. However, another p73 isoform, ΔNp73, which lacks the N-terminus, has oncogenic functions by interacting with and inhibiting the function of TAp73 and p53 [30]. The stagnant of p53-targeted drug development might be caused by the ignorance of the importance of p73 in cancer eradication [31]. Mutant p53 can also exert its gain of function by inhibiting p73 [28]. Several promising drugs that can repurpose p73, such as ALA-protoporphyrin IX and verteporfin, are under development [31]. In addition, in breast MDA-MB-231 cells, TGFβ and oncogenic RAS promote the interaction between mutant p53 and SMAD2 by triggering the phosphorylation of the mutant p53 N-terminus through CK1ɛ/δ kinases. Phosphorylated p53-R175H is able to bind to and inhibit tumor suppressor p63 with the help of SMAD2. p63 regulates many metastasis-related genes. Therefore, p53-R175H promotes metastasis by inhibiting the p63 pathway in breast cancer [32]. Additionally, p53-R175H promotes cancer-cell metastasis by activating EGFR/PIK3/AKT signaling pathway. Dong et al. observed that EGFR is activated after overexpressing p53-R175H in human endometrial-cancer cell line KLE, and is essential to its migration and invasion properties [33]. Muller et al. observed this phenomenon in breast cancer, and found that p53-R175H and p53-R273H enhance EGFR and integrin trafficking depending on the Rab-coupling protein (RCP), resulting in the activation of the EGFR and integrin pathway, and the promotion of cell migration and invasion [34]. Moreover, Zhang et al. discovered that RCP upregulated by mutant p53s is critical to the exosome-mediated Hsp90a secretion and causes a more invasive phenotype [35]. p53-R175H also affects the microRNA expression of cancer cells. p53-R175H, p53-R273H, and p53- C135Y directly bind to the promoter of miR-130b and suppress its expression. miR-130b negatively regulates the epithelial-to-mesenchymal transition (EMT) promoting gene *ZEB1*. Therefore, p53-R175H exerts its prometastatic ability through the miR-130b–ZEB1 axis in endometrial-cancer cells [36]. p53-R175H also inhibits miR-142-3p, which leads to PDAC cell progression and metastasis. p53-R175H transcriptionally upregulates DNA methyltransferase 1 (Dnmt1), resulting in the hypermethylation of the CpG islands of miR-142-3p, which is associated with tumor progression and metastasis [37]. In breast cancer, p53-R175H, p53-R273H, or p53-R280K interacts with HIF-1α, and transcriptionally upregulates miR-30d, which causes the enhancement of vesicular trafficking and secretion, leading to the remodeling of the extracellular matrix and the establishment of a microenvironment suitable for tumor growth and metastasis [38]. These findings suggest that p53-R175H-regulated microRNAs play roles in tumor metastasis. In prostate cancer, both inactivated p53 and p53-R175H upregulate cell-cycle progression genes. However, p53-R175H additionally upregulates developmental genes, including EMT regulator gene *TWIST1*. p53-R175H increases the expression of *TWIST1* by downregulating BMI-1, which maintains the level of epigenetic repression marker H3K27me3, causing the epithelial–mesenchymal transition and invasion of prostate-cancer cells [39]. Yeudall et al. reported that, in several cancer types, p53-R175H, p53-R273H, and p53-R281G contribute to cancer-cell migration by increasing the expression of several CXC-chemokines, including CXCL5, CXCL8, and CXCL12 [40]. In colorectal cancer, p53-R175H, p53-R248W, and p53-R273H can bind to SUMO-specific protease 1 (SENP1) and inhibit its de-SUMOylation function, causing the accumulation of SUMOylated Rac1, which is an activated form of RAC1 that promotes tumor growth and metastasis [41] (Table 4).

### 2.4. Promoting Tumor Initiation and Conferring Stemness

Since *TP53* mutations act as a driver in many cancer types [42], it is very likely that p53-R175H gains function in inducing tumorigenesis. Sarig et al. found that the reprogramming process of mouse embryonic fibroblasts (MEFs) expressing p53-R172H is enhanced comparing with p53 knockout MEFs. In addition, reprogrammed p53-R172H MEFs lost differentiation capacity and gained tumor initiation capacity [43]. Grespi et al. further found that p53 wild-type MEFs and p53-R172H MEFs have different expressions of microRNAs, which are associated with stemness, differentiation, and tumor suppression, implying that p53-R172H MEFs might gain tumor initiation capacity by regulating several microRNAs [44]. In the breast-cancer transgenic mouse model, p53-R175H/mWNT mice tend to develop multiple tumors, while most p53 ^−/−^/mWNT mice develop one tumor. p53-R175H expands mammary epithelial stem cells (MESCs) through the inactivation of ATM and causes tumorigenesis [45]. Expressing p53-R175H in human astrocytes also enhances its neurosphere-forming ability and increases cancer-stem-cell-like marker expression. On the basis of previous studies, p53-R175H can activate the EGFR integrin/PI3K/AKT pathway and promote tumor proliferation and metastasis [24,32,33]. In human astrocytes overexpressing p53-R175H, AKT2 that is activated by p53-R175H phosphorylates the WASP interacting protein (WIP), and the phosphorylated WIP stabilizes the YAP/TAZ complex, causing the transcriptional activation of genes that promotes cancer-stem-cell survival, including CYR61, CTGF, BIRC5 (also known as SURVIVIN), and CAV2 [46]. p53-R175H, p53-R248W, and p53-R273H increase the expression of colorectal cancer stem-cell markers CD44, LGR5, and ALDH in several colon cancer cell lines, and promote their tumor initiation capacity. Colon-cancer cells with high ALDH1 expression are also more resistant to cisplatin [47]. p53-R175H confers sphere-formation capacity on pancreatic-cancer cell line HPNE with KRAS-G12D mutation. In HPNE cells, p53-R175H induces the expression of HSP70, which binds to p53-R175H and maintains its stability [48]. Several p53 hotspot mutations promote hematopoietic stem- and progenitor cell (HSPC) expansion, and resistance to irradiation. p53-R175H, p53-R248W, and p53-R273H can directly bind to EZH2 and recruit EZH2 to chromatin, resulting in a global increase in H3K27me3 level and driving leukemia development [49]. p53-R175H, p53-R248W, and p53-R273H also upregulate EZH2 by repressing miR-26a in endometrial carcinoma [50]. However, p53-R175H acts as a tumor suppressor in certain conditions. Kadosh et al. discovered that, in the proximal gut, p53-R175H and p53-R273H inhibited the WNT signaling pathway by preventing the binding of transcription factor TCF4 to β-catenin. Interestingly, in the distal gut, gallic acid secreted by gut microbes abolished the effect of mutant p53s, thus turning on the WNT pathway and driving tumor initiation [51]. During tumor metastasis, cancer stem cells play an important role because they have self-renewal and tumor-initiation abilities. Increasing evidence shows that the EMT program is associated with the stemness program in cancer cells [52]. For example, in endometrial cancer, p53-R175H reduces miR-130b expression by directly binding to its promoter, thus inhibiting the expression of miR-130b target gene *ZEB1*, which functions as a transcriptional factor that is critical to cell EMT and stemness. ZEB1 can promote EMT by upregulating SNAI1, which can suppress the expression of epithelial marker E-cadherin. ZEB1 also promotes stemness by upregulating BMI-1, which is a chromatin remodeler that can reduce the expression of stemness-related transcription factors NANOG and KLF4 [36]. Moreover, ZEB1 upregulates chemoresistance-related genes *MDR1* and *MRP1* [36], in agreement with the concept that cancer stem cells are more resistant to chemotherapy [53] (Table 5). 

### 2.5. Promoting Tumor Drug Resistance, Inflammation, Angiogenesis, and Metabolic Reprogramming

Cancer stem cells are both responsible for tumor initiation, metastasis, and drug resistance [54], and associated with angiogenesis property [55] and metabolic reprogramming [56]. Several studies reported that many chemotherapy treatments have progressively become ineffective for patients with *TP53* mutation [57]. The inflammatory microenvironment can also confer cancer cells stemness and EMT [58]. It is very likely that mutant p53s are involved in these biological processes. Scian et al. overexpressed p53-R175H, p53-R273H, and p53-D281G in p53-null human lung-carcinoma cells H1299 and found that they are more resistant to common chemotherapy drug etoposide. These mutant p53s can bind to the promoter of the *NF-κB2* gene and increase its expression. NF-κB is involved in the antiapoptotic activity of cancer cells. Targeting the *NF-κB2* gene by siRNA in H1299 expressing mutant p53 can restore its sensitivity towards etoposide [59]. Donzelli et al. also found that. in lung-carcinoma cell lines H1299 and A549, p53-R175H enhanced the expression of miR128-2 through binding to the promoter of its host gene *ARPP21*. miR128-2 inhibited E2F5 expression and thus decreased the expression of E2F5 target gene *p21*. In cells overexpressing p53-R175H, p21 localized to the cytoplasmic compartment and exerted antiapoptotic function through preventing cleavage of procaspase-3 under a DNA damage condition. Therefore, p53-R175H confers lung-carcinoma cell resistance towards chemotherapy drugs doxorubicin, cisplatin, and 5-fluorouracil through the miR128-2/E2F5/p21 axis [60]. p53-R175H also exerts drug-resistance function through the proteasome degradation pathway. Wild-type p53 binds to the promoter of proteasome activator gene *REGγ* and recruits corepressor SMAD3 to repress its expression. p53-R175H also binds to the promoter of REGγ but prevents SMAD3 binding, thus upregulating *REGγ* expression and causing the degradation of several tumor suppressors including p53, p21, and p16, leading to cell proliferation and drug resistance [61]. *SLC25A1* is another gene transcriptionally activated by mutant p53s. Upregulation of SLC25A1 in the lung- or breast-cancer cells can cause resistance to platinum-based chemotherapy drugs [62]. In colon-cancer cells with DNA damage triggered by chemotherapy drugs, p53-R175H, p53-R248W, and p53-R273H upregulate ephrin-B2 coding gene *EFNB2* by binding to its promoter companies by NF-Y and p300. Upregulated ephrin-B2 promotes resistance to 5-FU by increasing resistance-associated gene *ABCG* expression through SRC/FAK/MEKK/MEK/JNK/c-JUN phosphorylation cascade. Ephrin-B2 induced by mutant p53 also promotes EMT, tumorigenesis, and proliferation through different SRC downstream phosphorylation cascades, emphasizing the important role of mutant p53 in tumor development [63]. 

p53-R175H also promotes angiogenesis. In breast cancer, p53-R175H and E2F1 bind to the promoter of *ID4* and enhance its expression. ID4 further binds to and stabilizes the mRNAs of IL8 and GRO-α, which are proangiogenic factors [64]. In 2005, it was reported that mutant p53s could upregulate the *NF-κB2* gene [59]. Transcription factor NF-κB plays a critical role in inflammatory responses and cancer development [65]. Weisz et al. reported that in the lung- and breast-cancer cell lines, p53-R175H can augment the induction of NF-κB expression in response to TNFα, which is a cytokine presented in a chronic inflammation environment, thereby promoting cancer progression. Cooks et al. further discovered that p53-R175H prolongs the activation period of NF-κB triggered by TNFα. Therefore, mice expressing p53-R175H are prone to develop inflammation-associated colon cancer [66]. p53-R175H can also promote the generation of inflammatory tumor microenvironment by regulating the secreted interleukin-1 receptor antagonist (sIL-1Ra). Mutant p53 but not wild-type p53 binds to the promoter of sIL-1Ra with corepressor MAFF and suppresses *sIL-1Ra* expression. Therefore, proinflammation cytokine IL-1 beta is not antagonized by sIL-1Ra [67]. Mutant p53 can also alter the metabolism of cancer cells. p53-R175H, p53-R248Q, and p53-R273H can induce the translocation of GLUT1 to the cell membrane through increasing the expression of small GTPases RhoA and ROCK, resulting in the stimulation of the Warburg effect, which means that cancer cells generate energy by using aerobic glycolysis instead of oxidative phosphorylation [68]. In head and neck cancer, under glucose deprivation conditions, p53-P151S, p53-R175H, p53-G245C, and p53-R282W localize to the cytoplasm and inhibit the phosphorylation and activation of AMPK, which is an energy sensor and can repress aerobic glycolysis. Therefore, mutant p53s induce the Warburg effect to promote tumor growth by inhibiting AMPK phosphorylation [69] (Table 6).

## 3. Targeting p53-R175H for Cancer Therapy

Because p53 is a very common mutated gene across many cancer types, scientists have attempted to develop drugs or strategies to target mutant p53 for a long time. There are three kinds of strategies which are reactivating the wild-type p53 function by promoting proper folding and stabilization of p53 mutants, promoting mutant p53 degradation, and immunotherapies based on mutant p53 neoantigen recognition. In this chapter, we summarize the current status of p53-R175H-targeted drugs development (Table 7).

### 3.1. Reactivating Wild-Type p53 Function

Many p53 mutations are missense mutations, generating proteins that only have one different amino acid from the wild-type p53 protein. Therefore, reactivating mutant p53 might be a strategy for cancer therapy. Scientists found that many p53 missense mutations can increase the free energy of the protein and decrease structural stability [84]. In the past two decades, many research groups put much effort into finding small molecules that can bind to and stabilize the p53 mutants, thereby reactivating their normal function. By different kinds of screening strategies such as protein screening, cellular screening, or molecular modeling, several small molecules have been identified that can restore the wild-type p53 function. Among them, there is a large group of molecules targeting the cysteine residues in p53 protein. These cysteine-binding compounds are soft electrophiles and can conjugate to the thiol group of the cysteine, which is a nucleophile, in mutant p53 proteins. This kind of chemical reaction is called the Michael addition, and these cysteine-binding compounds are referred to as Michael acceptors. PRIMA-1, which is a quinuclidinone, was identified by cellular screening on the basis of cancer-cell lines carrying tetracycline-regulated mutant p53. PRIMA-1 can suppress the growth of several types of tumor cells carrying p53-R175H and p53-R273H, but cannot inhibit the growth of tumor cells carrying wild-type p53 [70]. APR-246, a methylated form of PRIMA-1, is more active than PRIMA-1 and has a synergistic effect with cisplatin to inhibit the tumor xenograft growth in SCID mice [71]. Lambert et al. further discovered that both PRIMA-1 and APR-246 are converted into methylene quinuclidinone (MQ), which is a Michael acceptor that can conjugate to thiol groups in mutant *p53* proteins, while N-acetylcysteine (NAC) completely blocked its function. MIRA-1 [72] and STIMA-1 [73], which are p53-R175H and p53-R273H reactivators, also have reactive groups similar to PRIMA-1 and APR-246 that can act as Michael acceptors [85]. Wassman et al. further used in silico docking to find that PRIMA-1, APR-246, MIRA-1, and STIMA-1 all bind to the L1/S3 pocket of p53-R175H, which contains Cys124, Cys135, and Cys141 residues. The mutation of Cys124 can abolish the p53 reactivating effect of these compounds. They also identified stictic acid as a p53 reactivator by ensemble-based virtual screening against the L1/S3 pocket of the mutant p53. Treating cancer cell lines containing p53-R175H with stictic acid can increase the expression of the wild-type p53 target gene *p21* [74]. Peng et al. found that PRIMA-1 and APR-246 reactivate the mutant *p53* and inhibit thioredoxin reductase 1 (TrxR1), likely by Michael addition to the selenocysteine residue (U498), and convert its function into an NADPH oxidase, thereby increasing oxidative stress [86]. Tumor cells expressing mutant *p53* are more sensitive to reactive oxygen species (ROS) because of their aberrant metabolic pathways. For example, mutant p53 can downregulate AMPK, thereby reducing the activity of its downstream molecules PGC-1α and UCP2, which are involved in ROS scavenging [69]. Disrupting the REDOX balance might be a strategy to treat tumors containing mutant p53 [87]. APR-246, which can both reactivate mutant p53 proteins and increase the level of intracellular ROS, is a promising drug for p53-mutant tumors. Several ongoing clinical trials are testing the safety and efficacy of APR-246 for different cancer types (NCT03931291, NCT04214860, NCT04383938, NCT03745716, NCT04419389, NCT03588078, and NCT03072043 at ClinicalTrials.gov (accessed on 1 July 2021)). The latest result of the phase 2 trial (NCT03072043) showed combination treatment with eprenetapopt (APR-246) and azacitidine is well tolerated yielding high rates of clinical response and molecular remissions in patients with *TP53*-mutant myelodysplastic syndromes [88,89]. KSS-9 is a piperlongumine derivative with the insertion of combretastatin A4 at the C-7 position. It retained the ROS elevation function of piperlongumine and the microtubule disruption function of combretastatin A4. Surprisingly, KSS-9 also showed capacity to reactivate p53-R175H by performing Michael addition to its cysteine residues [75]. Recently, it has been reported that the cysteine residues of p53 mutants are reactive to arsenic. Chen et al. found that most structural p53 mutants can be stabilized by arsenic trioxide (ATO), and some of them are transcriptionally rescued, while DNA contact p53 mutants cannot be rescued by ATO. Crystal structures of p53 mutants incubated with AsI3 revealed that the arsenic atom is coordinated by Cys124, Cys135, and Cys141. This is an inspiring discovery because ATO was used to treat acute promyelocytic leukemia, and the result of in vitro and in vivo experiments showed that ATO suppressed tumor progression by reactivating many kinds of structural p53 mutants [76]. Several ongoing clinical trials are testing the safety and the efficacy of arsenic trioxide for acute myeloid leukemia, myelodysplastic syndromes, and refractory tumor with *TP53* mutation (NCT03931291, NCT04214860, NCT04383938, NCT03745716, NCT04419389, NCT03588078, and NCT03072043 at ClinicalTrials.gov (accessed on 1 July 2021)).

In addition to targeting cysteine residues of p53 mutants, there are other strategies to reactivate p53 mutants. ZMC1, which is a kind of thiosemicarbazone, was identified by screening NCI database substances that inhibit the growth of cancer cell lines expressing p53 hotspot mutations compared to that expressing wild-type p53. ZMC1 as a Zn^2+^ chelator can buffer the intracellular Zn^2+^ concentration, thereby improving the binding of Zn^2+^ specifically to p53-R175H [77]. It was known that the binding of Zn^2+^ is very important to the stability of wild-type p53 [8]. R175 residue lies closely to the Zn^2+^ binding site and the R175H mutation distorts the local structure, thereby affecting the Zn^2+^ binding ability [90]. ZMC1 reactivates p53-R175H and induces oxidative stress, likely by the ability of thiosemicarbazones to create hydroxyl radicals through the Fenton reaction [77]. Another thiosemicarbazone, COTI-2, can also reactivate p53 mutants and inhibit the PI3K-AKT pathway [91]. Some types of p53 mutants can form aggregates. Soragni et al. designed a peptide called ReACp53 that inhibits p53-R175H or p53-R248Q aggregate formation by targeting p53 segments 252-LTIITLE-258, which are essential for p53 aggregation. The inhibition of aggregation by ReACp53 reactivates p53 mutants and induces cell death and cell-cycle arrest in high-grade serous ovarian carcinomas [78]. The stability of p53 can be regulated by several heat-shock proteins or chaperones [92]. Hiraki et al. identified a small molecule called chetomin that can bind to Hsp40 and enhance the binding capacity of Hsp40 to p53-R175H, resulting in the conformational change of p53-R175H into a wild-type-like structure [93]. 

### 3.2. Promoting Mutant p53 Degradation

Promoting mutant *p53* degradation is another strategy to treat *TP53*-mutated cancers. The intracellular level of mutant p53 is mediated by DNAJA1, a member of the Hsp40 family. Statins were identified by cell-based screening as a compound that can degrade p53-R175H. Statins inhibit the mevalonate pathway. Mevalonate-5-phosphate (MVP), a metabolic intermediate in the mevalonate pathway, is reduced, thereby inhibiting the binding between p53-R175H and DNAJA1. p53-R175H, without the protection of DNAJA1, is led to proteasomal degradation by Hsc70-interacting protein (CHIP) mediated ubiquitination [79]. Mutant p53 can be stabilized by HDAC6-mediated Hsp90 deacetylation [81]. HDAC6/Hsp90-dependent mutant p53 accumulation is dependent on RhoA geranylgeranylation, which is downstream of the mevalonate pathway and can be inhibited by statins [80]. Since statins are commonly used lipid-lowering agents, they are promising drugs that target the mutant p53. Currently, several ongoing clinical trials are testing the efficacy of atorvastatin on triple-negative breast cancer, acute myeloid leukemia, and colorectal carcinoma (NCT03358017, NCT03560882, and NCT04767984 at ClinicalTrials.gov (accessed on 1 July 2021)). Targeting HDAC6 and Hsp90 is another way to induce the degradation of p53 mutants. Li et al. reported that the FDA-approved HDAC inhibitor suberoylanilide hydroxamic acid (SAHA) downregulated mutant p53 but not wild-type p53 in protein level, and showed cytotoxicity to breast-cancer cell lines expressing p53 mutants [81]. Furthermore, the treatment that combines SAHA with Hsp90 inhibitor 17DMAG inhibited T-cell lymphomagenesis in p53^R172H/R172H^ mice, and so did the Hsp90 inhibitor ganetespib alone treatment [82]. 

### 3.3. Immunotherapy Based on Mutant p53 Neoantigen Recognition

Since *TP53* missense mutations encode mutant p53 proteins that have one amino acid different from wild-type p53, scientists have long considered whether these proteins can be presented by tumor cells or antigen-presenting cells, and can be a target for immunotherapy. In 2018, Hwang et al. successfully generated monoclonal antibodies against p53-R175H, p53-R248Q, and p53-R273H with high specificity by designing immunogens that are TrxA proteins fused with three copies of the mutant p53 sequence. These antibodies can be used in the clinic to detect the p53 status rapidly [94]. Lo et al. reported that some tumor-infiltrating lymphocytes (TILs) isolated from a patient with metastatic colorectal cancer were able to recognize p53-R175H restricted to HLA-A*0201. HLA-A*0201 is the most common allele in HLA-A2 subtypes, especially among general Caucasian populations [95]. Three T-cell receptors (TCRs) were identified that can recognize cell lines expressing both p53-R175H and HLA-A*0201 [96]. To understand whether the recognition of *p53* mutants by the TILs is a common phenomenon across different cancer types, TILs from 28 patients with several types of cancers were screened by autologous antigen-presenting cells that present corresponding p53 mutant peptides by electroporated with tandem minigenes or pulsed with peptides. TILs from 39% of patients were able to recognize mutant p53 peptides restricted to HLA class I or class II and be activated. This study suggested that p53 mutants could be a target for cancer immunotherapy [97]. Furthermore, peripheral blood lymphocytes (PBLs) from patients whose TILs were reactive to mutated p53 could also respond to antigen-presenting cells presenting p53 mutant peptides. PBLs can be obtained by non-invasive methods, making them a more accessible source of T cells for adoptive cell therapy [98]. Wu et al. deeply examined the structure of the TCR–p53R175H–HLA-A2 complex to understand how p53R175H-reactive TCRs distinguish p53-R175H from the wild-type and found that TCRs were shifted towards the C-terminus of the p53-R175H peptide. This knowledge could contribute to the future design of TCRs against p53 mutants [99]. Recently, Hsiue et al. made a big step in the development of mutant p53 targeting immunotherapy by constructing bispecific antibody H2-scDb that could bind to the p53-R175H peptide-HLA-A*0201 complex with one arm and the TCR-CD3 with the other arm. Although the density of the p53-R175H peptide-HLA-A*0201 complex in the surface of tumor cells is very low, bispecific antibody H2-scDb activated T cells to kill tumor cells in vitro and in mice [83]. Mutant p53 peptide neoantigens are a promising target for cancer immunotherapy because of the high occurrence of *TP53* missense mutation across cancer types. 

## 4. Conclusions

By summarizing the p53-R175H gain-of-functions reported by different groups, p53-R175H and other p53 hotspot mutations were found to contribute to tumor development by promoting cancer cell proliferation, migration, invasion, initiation, metabolic reprogramming, angiogenesis, and conferring drug resistance to cancer cells (Table 2). The effect of mutant p53 in cancer cells is so broad and huge, giving it the title of “the guardian of the cancer cell” [100]. Furthermore, p53-R175H often executes its function by directly binding to promoters of genes and transcriptionally up- or down-regulating their expressions by recruiting cofactors or corepressors, respectively. p53-R175H interacts with other transcription factors such as p63, p53, ETS2, and NF-Y to interfere in the expression of their target genes. Different kinds of p53 mutants have unique functions. However, R273H, R248W, and R248Q have similar functions with those of R175H. The function of p53-R175H is highly dependent on the cellular context. p53-R175H influences different kinds of pathways in different kinds of cancer types. Currently, the p53-R175H gain-of-function study is intensive on several cancer types, including breast, lung, colorectal, and pancreatic cancers. However, *TP53* is also highly mutated in other cancer types such as osteosarcoma, and ovarian, head and neck, esophageal, and brain cancers. More p53 gain-of-function studies are needed in these types of cancers in order to find the vulnerabilities of these cancer cells. Regarding treatment targeting p53-R175H, many small molecules were identified by screening that could reactivate p53 tumor-suppressor activities. Several small molecules were also found to promote the degradation of p53-R175H mutants by targeting heat-shock proteins that bind and stabilize p53-R175H. Recently, it was discovered that p53-R175H peptides-HLA complexes are presented on the cell surface of cancer cells, and T cells can distinguish them from the wild-type p53 peptide, making it become a promising target for cancer immunotherapies. 

## Figures and Tables

**Figure 1 cancers-13-04088-f001:**
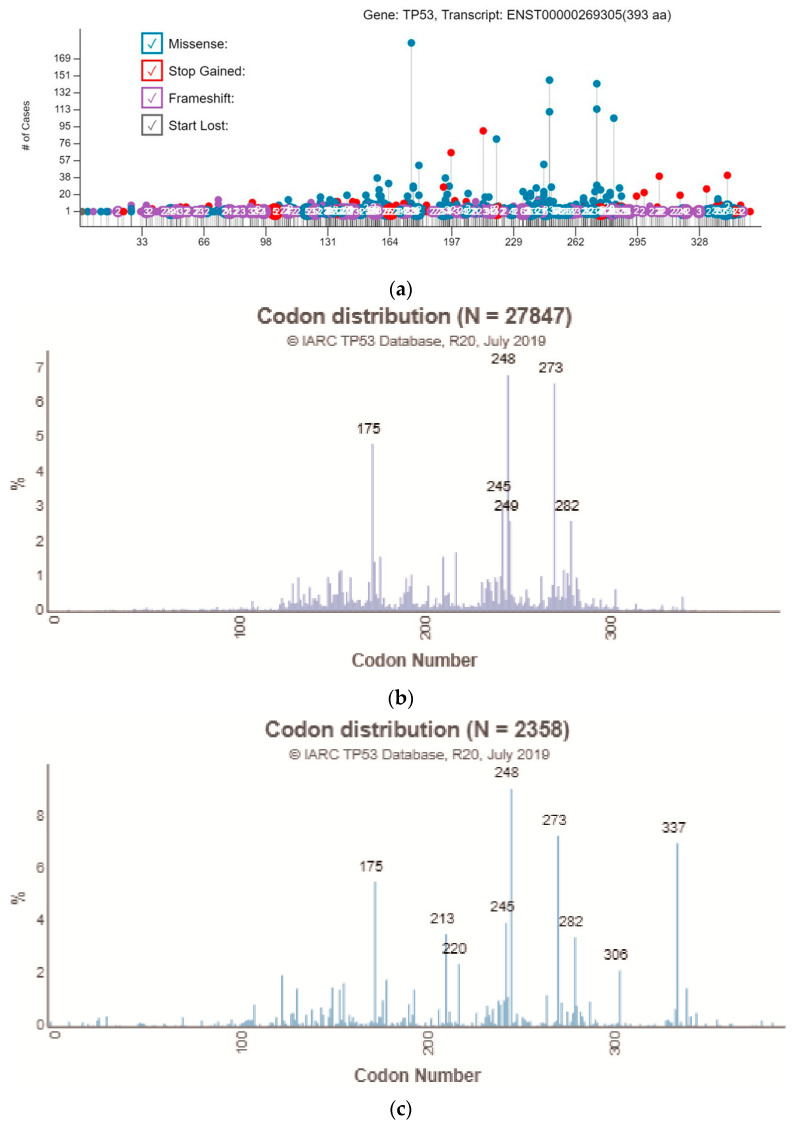
Distribution of *TP53* mutations. (**a**) Distribution of *TP53* mutations from the TCGA database. Picture obtained from National Cancer Institute GDC Data Portal; (**b**) Distribution of TP53 somatic mutations from the IARC TP53 database; (**c**) Distribution of *TP53* germline mutations from the IARC TP53 database; and (**b**,**c**) Picture obtained from the IARC TP53 database website (https://p53.iarc.fr/ (accessed on 1 July 2021)).

**Table 1 cancers-13-04088-t001:** Top 10 frequently occurring *TP53* mutants. Data were obtained from the TCGA database and the IARC *TP53* somatic mutations database.

TCGA Database		IARC *TP53* Somatic Mutations Database	
Mutants	Percentage in *TP53* Mutant Cases	Mutants	Percentage in *TP53* Mutant Cases
Missense *TP53* R175H	3.69%	Missense R175H	4.21%
Missense *TP53* R248Q	2.88%	Missense R248Q	3.28%
Missense *TP53* R273C	2.80%	Missense R273H	2.97%
Missense *TP53* R273H	2.25%	Missense R248W	2.65%
Missense *TP53* R248W	2.19%	Missense R273C	2.45%
Missense *TP53* R282W	2.05%	Missense R282W	2.10%
Stop Gained *TP53* R213 *	1.78%	Missense G245S	1.58%
Missense *TP53* Y220C	1.60%	Missense R249S	1.52%
Stop Gained *TP53* R196 *	1.30%	Missense Y220C	1.39%
Missense *TP53* G245S	1.05%	Stop Gained TP53 R213 ^1^ *	1.14%

^1^ * represents the termination codon.

**Table 2 cancers-13-04088-t002:** Summary of p53-R175H gain-of-function of inducing genetic instability.

Tumor Type	Same Function Mutations	R175h Interacting Protein	Affected Downstream Molecule or Pathway	Ref.
Pancreatic cancer	-	-	-	[16]
None (mouse embryonic fibroblast)	R248W, R273H	Mre11	MRN/ATM	[17]

**Table 3 cancers-13-04088-t003:** Summary of p53-R175H gain-of-function of promoting cancer cell growth.

Tumor Type	Same Function Mutations	R175H Interacting Protein	Affected Downstream Molecule or Pathway	Ref.
Breast cancer	R248Q, R248W, R249S, R273H	ETS2	MLL1, MLL2, MOZ/Global chromatin remodeling	[21]
Breast cancer	R248Q, R273H, R280K	NF-Y, YAP	*Cyclin A*, *cyclin B*, and *CDK1* genes	[22]
Pancreatic cancer	-	-	hnRNPK/GAP17 isoforms/KRAS signaling pathway	[25]
Bladder, bone, and ovarian cancer	V157F, 248Q	-	GEF-H1	[19]
Colorectal and pancreatic cancer	-	-	GRO1	[20]
Breast, colorectal, and lung cancer	R273H	-	CDC7, Dbf4	[23]
Breast and prostate cancer	R280K	DAB2IP	PI3K/AKT1	[24]

**Table 4 cancers-13-04088-t004:** Summary of p53-R175H gain-of-function of promoting tumor migration, invasion, and metastasis.

Tumor Type	Same Function Mutations	R175H Interacting Protein	Affected Downstream Molecule or Pathway	Ref.
Breast cancer	-	p63, Smad2	p63 target genes	[32]
Breast cancer	R273H	-	RCP/EGFR and integrin/PI3K/AKT	[34]
Breast cancer	R273H	-	RCP/Hsp90a secretion	[35]
Breast cancer	R273H, R280K	HIF-1α	miR-30d/accelerated vesicular trafficking	[38]
Colorectal cancer	R248W, R273H	SENP1	Activation of Rac1 by SUMOylation	[41]
Endometrial cancer	R273H, C135Y	-	miR-130b/ZEB1/Snail/E-cad	[36]
Pancreatic cancer	-	p73	NF-YA, NF-YB, and NF-YC complex/PDGFRb	[28]
Pancreatic cancer	-	-	Dnmt1/MiR-142-3p	[37]
Prostate cancer	-	-	BMI-1/Twist1	[39]
Breast, lung, and skin cancer	R273H, D281G	-	CXCL5, CXCL8, and CXCL12	[40]

**Table 5 cancers-13-04088-t005:** Summary of p53-R175H gain-of-function promoting tumor initiation and conferring stemness.

Tumor Type	Same Function Mutations	R175H Interacting Protein	Affected Downstream Molecule or Pathway	Ref.
Colorectal cancer	R248W, R273H	-	Stem cell markers CD44, LGR5, and ALDH1	[47]
Colorectal cancer	R273H	TCF4	β-catenin/WNT signaling pathway	[51]
Leukemia	R248W, R273H	EZH2	Global increase of H3K27me3	[49]
Brain and breast cancer	R273H	-	RCP/EGFR and integrin/PI3K/AKT/WIP/YAP, TAZ	[46]

**Table 6 cancers-13-04088-t006:** Summary of p53-R175H gain-of-function of promoting tumor drug-resistance, inflammation, angiogenesis, and metabolic reprogramming.

Gain-of-Function	Tumor Type	Same Function Mutations	R175H Interacting Protein	Affected Downstream Molecule or Pathway	Ref.
Chemoresistance	Colorectal cancer	R273H, R248W	NF-Y, p300	Ephrin-B2/SRC/FAK/ MEKK/MEK/JNK/c-JUN/ABCG	[63]
	Lung cancer	R273H, D281G	-	NF-κB2	[59]
	Lung cancer	-	-	miR128-2/E2F5/p21/pro-caspase-3	[60]
	Breast, colorectal, head and neck, and lung cancer	R248W, R273H, R280K, R282W	SMDA3	REGγ/proteasome activation/p53, p21, and p16 degradation	[61]
	Breast and lung cancer	G245A, D281G, R273H, R280K	-	SLC25A1	[62]
Angiogenesis	Breast cancer	R273H, R280K	E2F1	ID4/IL8, GRO-α	[64]
Inflammatory response	Colorectal cancer	R273H	-	*NF-κB* expression in response to TNFα	[66]
	Breast, colorectal, and liver cancer	R273H, R280K	MAFF	sIL-1Ra/IL-1β	[67]
Metabolic reprogramming	Breast and lung cancer	R248Q, R273H	-	RhoA, ROCK/GLUT1 translocation/Warburg effect	[68]
	Head and neck cancer	P151S, G245C, R282W	AMPK	AMPK inactivation/Warburg effect	[69]

**Table 7 cancers-13-04088-t007:** Summary of drugs and strategies targeting p53-R175H for cancer therapy.

Concept	Strategy	Drug	Other Targeted *p53* Mutants	Clinical Trial	Phase	Status	Ref.
Reactivating wild-type p53 function	targeting the cysteine residues of p53	PRIMA-1	R273H	-	-	-	[70]
		APR-246	R273H	NCT03931291	2	Active, not recruiting	[71]
				NCT04214860	1	Recruiting	
				NCT04383938	1, 2	Recruiting	
				NCT03745716	3	Active, not recruiting	
				NCT04419389	1, 2	Recruiting	
				NCT03588078	1, 2	Active, not recruiting	
				NCT03072043	1, 2	Active, not recruiting	
		MIRA-1	R273H	-	-	-	[72]
		STIMA-1	R273H	-	-	-	[73]
		Stictic acid	-	-	-	-	[74]
		KSS-9	-	-	-	-	[75]
		Arsenic trioxide	Structural *p53* mutants	NCT04695223	2	Recruiting	[76]
				NCT03855371	1	Recruiting	
				NCT03377725	3	unknown	
				NCT04869475	2	Recruiting	
	Other strategies	ZMC1	-	-	-	-	[77]
		ReACp53	R248Q	-	-	-	[78]
Promoting mutant *p53* degradation	Targeting the mevalonate pathway	Atorvastatin	Structural *p53* mutants	NCT03560882	1	Recruiting	[79,80]
				NCT04767984	2	Not yet recruiting	
				NCT03358017	2	Recruiting	
	HDAC inhibitor	SAHA	Broad effect	NCT02042989	1	Active, not recruiting	[81]
	Hsp90 inhibitor	Ganetespib	Broad effect	-	-	-	[82]
Immuno- therapy	Bispecific antibody	H2-scDb	-	-	-	-	[83]

## Data Availability

Publicly available datasets were analyzed in this study. This data can be found here: [The TCGA database: (https://www.cancer.gov/tcga) (accessed on 1 July 2021). The IARC *TP53* database: (https://p53.iarc.fr/) (accessed on 1 July 2021)].

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
