# Peer review of "The Function of the Mutant p53-R175H in Cancer"

_cancers, 2021, doi:10.3390/cancers13164088_

Round 1
Reviewer 1 Report
The review “The function of mutant p53-R175H in cancer” by Chiang et al, provides a comprehensible archive of information on the highly frequent R175H mutation of the TP53 gene. The authors have done an excellent review of the topic and I have nothing to take away from this review. I believe this review will serve as an important point of reference for future studies. On that regard, I request the authors to incorporate some important studies on R175H mutation as follows, which should improve the quality of the review further:
- Liu et al, 2010, Oncogene, PMID: 19881536
- Li et al., 2020, Phys Chem Chem Phys, PMID: 32307496
- Stein et al., 2019, IJMS, PMID: 31817996
Author Response
Point-by-Point Response for Reviewer 1
Response: We are deeply appreciative of your friendly comments and suggestion. The suggestions have now been incorporated into our revised manuscript.
Point #1:
I believe this review will serve as an important point of reference for future studies. On that regard, I request the authors to incorporate some important studies on R175H mutation as follows, which should improve the quality of the review further:
- Liu et al, 2010, Oncogene, PMID: 19881536
- Li et al., 2020, Phys Chem Chem Phys, PMID: 32307496
- Stein et al., 2019, IJMS, PMID: 31817996
Response to Point#1
We have included these references in our review.
(Liu et al, 2010, Oncogene, PMID: 19881536): The study has been incorporated into lines 154-157 as a citation [16]. Furthermore, because of the importance of the p53-R175H gain-of-function in genetic instability, we added a new section “2.1. Inducing Genetic Instability” in lines 145-157 to introduce it.
(Li et al., 2020, Phys Chem Chem Phys, PMID: 32307496): The study has been incorporated into lines 100-102 as a citation [11] in the Introduction.
(Stein et al., 2019, IJMS, PMID: 31817996): The study has been incorporated into lines 100-102 as a citation [10] in the Introduction.
Reviewer 2 Report
The work by Chiang et al. strives to describe the function of p53-R175H mutant protein in cancers.
The reviewer agrees, that it might be helpful to summarize the up-to-date information on certain p53 mutants, yet, the information included in the submitted work must be carefully revised before the work can be considered for publication in Cancers.
First, the authors should explain why the focus is on R175H mutation. According to common knowledge, this is a hotspot mutation that renders p53 unfolded and results in the loss of DNA binding capacity and protein aggregation. This should be clearly emphasized in the abstract and through the text.
I agree that the general information on cancer in the Introduction might be relevant for a more general audience, yet the review should focus on p53 primarily. Authors should also refer to IARC TP53 somatic and germline mutation database while assessing the frequency of hotspots in TP53.
The manuscript should be re-organized in such a way it is easy for the reader to follow why certain mutp53 targets are listed (e.g. section 2.1. Promoting Tumor Cell Growth). The scheme or table summarizing each part would be of help. The abbreviations are rarely explained. This must be addressed.
Also, since the authors chose to focus on 175mutp53 solely, the reviewer is not convinced that adding the information on other mutp53 is helpful. I agree it might be relevant, yet the authors should reshuffle the information to enable the clarity of the message.
Regarding section 3 on targeting mutp53, the newest literature about the compounds and the outcomes of clinical trials must be included (eg for APR-246).
Authors should double-check the grammar and syntax.
Author Response
Point-by-Point Response for Reviewer 2
Response: We truly appreciate your valuable critiques and suggestions for our manuscript. The suggestions have been incorporated into our revised manuscript. We have made the point-by-point responses to the comments and suggestions as follows.
Point #1:
The authors should explain why the focus is on R175H mutation. According to common knowledge, this is a hotspot mutation that renders p53 unfolded and results in the loss of DNA binding capacity and protein aggregation. This should be clearly emphasized in the abstract and through the text.
Response to Point#1:
Thanks for the reviewer’s comments. We choose to focus on p53-R175H because among p53 hotspot mutations, p53-R175H has the highest occurrence in cancer patients in both the TCGA database and IARC TP53 somatic mutations database (Table 1). The gain-of-functions of p53-R175H have been reported by many research groups in different kinds of cancer types. The p53-R175H loses the transactivating function of the wild-type p53 due to its defective DNA binding interface. However, p53-R175H gains function by many kinds of mechanisms. It is reported that p53-R175H can bind to some DNA sequences which are different from the wild-type p53 response element and transactivate its target gene. p53-R175H can also interact with numerous kinds of transcription factors to enhance or suppress the expression of their target genes. In addition, the p53-R175H is more prone to aggregation than the wild-type p53 because it exhibits a larger hydrophobic surface area and higher loop flexibility than the wild-type p53. p53-R175H aggregates can not only induce coaggregation of wild-type p53 to cause the loss of function but also induce coaggregation of p63 and p73 to cause gain-of-functions. The explanation has been incorporated into lines 93-107 in the Introduction and into lines 39-40 in the Abstract.
Point #2:
I agree that the general information on cancer in the Introduction might be relevant for a more general audience, yet the review should focus on p53 primarily. Authors should also refer to IARC TP53 somatic and germline mutation database while assessing the frequency of hotspots in TP53.
Response to Point#2:
Thanks for the reviewer’s suggestion. The purpose of the general information on cancer in the Introduction is to let the general audience understand how p53 loss-of-function and gain-of-function cause tumorigenesis. We have made a more detailed description about p53 in lines 59-64 in the Introduction to make the Introduction focus on p53.
Point #3:
The manuscript should be re-organized in such a way it is easy for the reader to follow why certain mutp53 targets are listed (e.g. section 2.1. Promoting Tumor Cell Growth). The scheme or table summarizing each part would be of help. The abbreviations are rarely explained. This must be addressed.
Response to Point#3:
Based on the reviewer’s suggestions, we re-organized our manuscript. In order to make readers easy to follow why certain mutp53 targets are listed, we separated the original Table 2 into Table 2, 3, 4, 5, and 6 for each section in Chapter 2: “Gain of Function of Mutant p53-R175H”. We also add a section in lines 124-144 in Chapter 2 to make readers understand why we would like to discuss the gain-of-function of p53-R175H. Furthermore, we add a table (Table 7) in line 525 in Chapter 3: “Targeting p53-R175H for Cancer Therapy” to summarize drugs and strategies that target p53-R175H for cancer therapy. As for the abbreviations, most abbreviations are explained in the place where they appear the first time in the context.
Point #4:
Since the authors chose to focus on 175mutp53 solely, the reviewer is not convinced that adding the information on other mutp53 is helpful. I agree it might be relevant, yet the authors should reshuffle the information to enable the clarity of the message.
Response to Point#4:
We sincerely appreciate the reviewer’s constructive comment. We list other p53 mutants that have gain-of-function as same as p53-R175H in each study to make it easy to find which p53 mutants have phenotypes similar to p53-R175H or which molecular mechanisms are p53-R175H specific. The explanation has been incorporated into lines 141-144 in Chapter 2: “Gain of Function of Mutant p53-R175H”.
Point #5:
Regarding section 3 on targeting mutp53, the newest literature about the compounds and the outcomes of clinical trials must be included (e.g. for APR-246)
Response to Point#5:
Based on the reviewer’s suggestion, we have included the newest outcomes and literature about the compound APR-246 in lines 430-433 as citation [77,78] in section “3.1. Reactivating Wild-Type p53 Function”. The status and the phase of clinical trials for compounds mentioned in Chapter 3: “Targeting p53-R175H for Cancer Therapy” are all listed in Table 7 in line 525.
Point #6:
Authors should double-check the grammar and syntax.
Response to Point#6:
The manuscript has been double-checked by authors and undergone English language editing by a professional English editor (MDPI Editing number: English-32354).
Round 2
Reviewer 2 Report
The authors have addressed all comments.
A summary figure showing therapeutic strategies aiming at the restoration of p53-175 mutant would be of benefit.
Please double-check the resolution of the figure included and the reference style to the database.
Author Response
Point-by-Point Response
Response: We are deeply appreciative of your comments and suggestion. The suggestions have been incorporated into our revised manuscript. We have made the point-by-point responses to the comments and suggestions as follows.
Point #1:
A summary figure showing therapeutic strategies aiming at the restoration of p53-175 mutant would be of benefit.
Response to Point#1:
Thanks for the reviewer’s suggestion. A summary figure showing therapeutic strategies aiming at the restoration of p53-175 mutant has been previously incorporated into the Graphic Abstract as follows.
Point #2:
Please double-check the resolution of the figure included and the reference style to the database.
Response to Point#2:
Thanks for the reviewer’s suggestion. The resolution of the figure included have been increased. All of the figures are provided at a sufficiently high resolution (minimum 1000 pixels width/height, or a resolution of 300 dpi or higher).
As for the reference style to the database, we used data from the TCGA database and the IARC TP53 database. According to the request of the TCGA (https://www.cancer.gov/about-nci/organization/ccg/research/structural-genomics/tcga/using-tcga/citing-tcga), we acknowledged the TCGA Research Network in the acknowledgements section of our work as "The results shown here are in whole or part based upon data generated by the TCGA Research Network: https://www.cancer.gov/tcga.". According to the request of the TCGA IARC TP53 database (https://p53.iarc.fr/), we cited "Bouaoun L, Sonkin D, Ardin M, Hollstein M, Byrnes G, Zavadil J, Olivier M. TP53 Variations in Human Cancers: New Lessons from the IARC TP53 Database and Genomics Data (R20). Hum Mutat. 2016 Sep;37(9):865-76" as reference [4].
